# Synthesis and Structure of *Nido*-Carboranyl Azide and Its “Click” Reactions

**DOI:** 10.3390/molecules26030530

**Published:** 2021-01-20

**Authors:** Anna A. Druzina, Olga B. Zhidkova, Nadezhda V. Dudarova, Irina D. Kosenko, Ivan V. Ananyev, Sergey V. Timofeev, Vladimir I. Bregadze

**Affiliations:** A.N. Nesmeyanov Institute of Organoelement Compounds, Russian Academy of Sciences, 28 Vavilov Str., 119991 Moscow, Russia; Zolga57@mail.ru (O.B.Z.); nadezjdino_96@mail.ru (N.V.D.); kosenko@ineos.ac.ru (I.D.K.); i.ananyev@gmail.com (I.V.A.); timofeev@ineos.ac.ru (S.V.T.); bre@ineos.ac.ru (V.I.B.)

**Keywords:** *nido*-carborane, boronated azide, cobalt bis(dicarbollide), “click” reaction, cholesterol, X-ray diffraction

## Abstract

Novel zwitter-ionic *nido*-carboranyl azide 9-N_3_(CH_2_)_3_Me_2_N-*nido*-7,8-C_2_B_9_H_11_ was prepared by the reaction of 9-Cl(CH_2_)_3_Me_2_N-*nido*-7,8-C_2_B_9_H_11_ with NaN_3_. The solid-state molecular structure of *nido*-carboranyl azide was determined by single-crystal X-ray diffraction. 9-N_3_(CH_2_)_3_Me_2_N-*nido*-7,8-C_2_B_9_H_11_ was used for the copper(I)-catalyzed azide-alkyne cycloaddition with phenylacetylene, alkynyl-3β-cholesterol and cobalt/iron bis(dicarbollide) terminal alkynes to form the target 1,2,3-triazoles. The *nido*-carborane-cholesterol conjugate 9-3β-Chol-O(CH_2_)C-CH-N_3_(CH_2_)_3_Me_2_N-*nido*-7,8-C_2_B_9_H_11_ with charge-compensated group in a linker can be used as a precursor for preparation of liposomes for Boron Neutron Capture Therapy (BNCT). A series of novel zwitter-ionic boron-enriched cluster compounds bearing a 1,2,3-triazol-metallacarborane-carborane conjugated system was synthesized. Prepared conjugates contain a large amount of boron atom in the biomolecule and potentially can be used for BNCT.

## 1. Introduction

*Nido*-Carborane (7,8-dicarba-*nido*-undecaborate anion) and its derivatives attract attention due to its unique electronic structure, namely, electron delocalization, which is often considered as an unusual three-dimensional aromaticity [1,2], and also due to its electron-withdrawing character of the skeletal carbon atoms. Such structural and physicochemical properties of 7,8-dicarba-*nido*-undecaborate anion induce the preparation of new functional derivatives of *nido*-carboranes with a variety of practical applications. These compounds attract the continued interest of researchers working in various fields, such as medical chemistry (boron neutron capture therapy (BNCT) of malignant tumors [3,4,5], HIV protease inhibitors [6], reagents for radioimaging of tumors [7]), and creation of new materials [1] (carborane-containing polymers, ionic liquids, liquid crystals).

It is well known that BNCT is a binary method of cancer treatment, in which it is necessary that boron compounds selectively accumulate in the tumor tissue at the required therapeutic concentration for its subsequent irradiation with thermal neutrons [8,9,10]. The advantage of using derivatives of *nido*-carboranes in BNCT is their stability and a high content of boron atoms in the molecule. In this regard, the new methods for the synthesis of various functional mono-substituted derivatives of *nido*-carborane should be developed. Currently, there are several methods for the preparation of mono-substituted *nido*-carboranes: C-substituted derivatives can be obtained by modification of the carborane cage at carbon atoms of the parent *ortho*-carborane followed by its deboronation to the corresponding *nido*-carborane. Last time, the most cited methods of synthesis B-substituted functional derivatives of *nido*-carboranes are ring opening of their cyclic oxonium derivatives under the action of nucleophilic agents [11,12,13,14], alkylation of their methylsulfide derivatives [15,16,17,18], Cu-promoted synthesis of their ammonium derivatives [19] and nucleophilic addition of alcohols and mercaptans to highly polarized triple bond B-N^+^≡CR [20].

Synthesis of mono-substituted derivatives of *nido*-carborane with a functional group makes it possible to attach a carborane fragment to various bio- and macromolecules and thus to obtain compounds with a given set of properties. The substituent introduced can be a biologically active derivative that acts as a tumor targeting vector, or a simple functional group, which can be used for conjugation with high molecular weight biomolecules using standard methods of bioorganic chemistry. Recently the Cu(I)-catalyzed reaction of 1,3-dipolar [3 + 2]-cycloaddition of azides with alkynes (“click” reaction) has found more and more widespread use for the bioconjugation of molecules [21,22,23,24,25]. Such reactions must proceed rapidly under ambient conditions, resulting in a high yield of desired 1,2,3-triazole. The 1,2,3-triazole scaffold is known to be one of the most valuable in the chemistry of biologically active compounds [26], which exhibit anticancer [27,28,29], anti-HIV [27,29,30], antibacterial [27,29,31,32], antioxidant [27,29,33,34] activities. In addition, the 1,2,3-triazole in the molecules between boron-containing and the biologically active fragments (porphyrins, nucleosides) is a linking unit that mimics geometry and electronic properties of the peptide bond is a more stable to hydrolysis reactions [14,35]. However, the preparation of a suitable *nido*-carborane-containing substrate for a “click” reaction often requires multistage syntheses, for instance, either the incorporation of a distant or cage-bound azide group or a terminal alkyne. Azide is one the most popular biorthogonal functional groups due to its small size coupled with stability to water and inertness towards endogenous biological functionalities [36]. It should be noted that only a few examples of azido-containing *nido*-carboranes are known to date. Recently we have obtained C-substituted *nido*-carboranyl azides [7-N_3_CH_2_CH_2_OCH_2_CH_2_O-*nido*-7,8-C_2_B_9_H_11_]^−^ and [7-N_3_CH_2_CH_2_OCH_2_CH_2_S-*nido*-7,8-C_2_B_9_H_11_]^−^ which were synthesized by alkylation of 1-mercapto-*ortho*-carborane with bis(2-chloroethyl) ether followed by introduction of azide group and by the conversion of *closo*-derivative to *nido*-form [37]. B-substituted [10-N_3_CH_2_CH_2_CH_2_CH_2_O-*nido*-7,8-C_2_B_9_H_11_]^−^ and [10-N_3_CH_2_CH_2_OCH_2_CH_2_O-*nido*-7,8-C_2_B_9_H_11_]^−^ were prepared by the ring-opening reactions of the corresponding cyclic oxonium derivatives with sodium azide [11,14].

In this contribution, we describe a synthesis of zwitter-ionic B-substituted *nido*-carborane bearing a functional azido-group 9-N_3_(CH_2_)_3_Me_2_N-*nido*-7,8-C_2_B_9_H_11_ and study its behaviour in the copper(I)-catalyzed azide-alkyne cycloaddition reaction with phenylacetylene, alkynyl-3β-cholesterol and terminal alkynes derivatives of cobalt/iron bis(dicarbollide).

## 2. Results and Discussion

### 2.1. Synthesis of 9-N_3_(CH_2_)_3_Me_2_N-nido-7,8-C_2_B_9_H_11_ and Its “Click” Reaction with Phenylacetylene 

The reaction of 9-Cl(CH_2_)_3_Me_2_N-*nido*-7,8-C_2_B_9_H_11_
**1** with NaN_3_ in the presence of NaI as a catalyst in DMF upon prolonged heating under 50 °C for 7 days results in 9-N_3_(CH_2_)_3_Me_2_N-*nido*-7,8-C_2_B_9_H_11_
**2** with 90% yield (Scheme 1). It was isolated as a white non-hydroscopic solid soluble in common organic solvents like CH_2_Cl_2_, CH_3_CN, alcohols and non-soluble in hydrocarbons and water. 

The ^11^B-NMR spectrum contains eight signals indicating a nonsymmetrical monosubstituted structure. The ^1^H-NMR spectrum of the signal of the methylene group bonded to the nitrogen atom exhibits a singlet at 3.37 ppm. It should be noted that the splitting reduces as the distance from the nitrogen atom increases resulting in singlet and multiplet at 2.15 and 3.50 ppm for the second and third methylene groups. In addition, the signals of the C*H_carb_* groups and the *extra-*hydrogen are observed approx. at 2.59 and -3.4 ppm, correspondingly. In the ^13^C-NMR spectrum the most characteristic is signal of *C*H_2_N_3_ group. As observed earlier [19], the signal of the *C*H_2_Cl group in the ^13^C-NMR spectrum appears at 43.0 ppm for **1**. The substitution of chlorine for azide results in the low-field shift to 65.2 ppm. The azide stretching band in the IR spectrum of **2** is located at 2075 cm^-1^.

Conditions of “click” reactions in the preparation of various boron-containing biomolecules vary particularly wide. Earlier “click” reactions have been successfully used for the synthesis of conjugates of bis(dicarbollide) metallacarboranes and *nido*-carborane with thymidine [14]. The synthesis was carried out in a mixture of *tert*-butanol/water (1:1) at ambient temperature using copper(II) sulfate pentahydrate with potassium ascorbate as a catalyst. The same reaction for synthesis of conjugate dodecaborate dianion with thymidine proceed in CH_3_CN at ambient temperature with copper(II) sulfate pentahydrate with sodium ascorbate [38]. Conjugates of chlorine *e_6_* with a cobalt bis(dicarbollide) anion or *closo*-dodecaborate dianion were obtained using CuI and Et_3_N in acetonitrile at ambient temperatures [35]. Series of 1,2,3-triazoles bearing *closo*-dodecaborate fragment was obtained using CuI as a catalyst and Et_3_N as a base under reflux in ethanol [39].

In the present work, we studied the behavior of azide **2** in copper(I)-catalyzed azide-alkyne cycloaddition using simple terminal alkyne as a pilot compound. It was showed that it readily reacts with phenylacetylene in ethanol in the presence of diisopropylethylamine (DIPEA) and catalytic amount of CuI to give the corresponding 1,2,3-triazole **3** with 85 % yield (Scheme 2). 

The structure of the *nido*-carborane **3** was confirmed by the data of ^1^H-, ^11^B- and ^13^C-NMR, IR spectroscopy and HRMS. The ^1^H- and ^13^C-NMR spectra of compounds **3** along with the signals of the heteroaliphatic chain and the phenyl group contain the characteristic signals of the triazole cycle in the ^1^H-NMR spectrum signal of the C*H_triazole_* hydrogen appears at 8.62 ppm. In the ^13^C-NMR spectrum, the signal of the *C*H*_triazole_* carbon is observed at 122.0 ppm, whereas the signal of the *C_triazole_* carbon appears at 146.9 ppm. In the ^1^H-NMR spectrum, the signal of the methylene group next to the triazole cycle is observed at 4.54 ppm and the characteristic signal of the *Me_2_*N hydrogens appear at 2.88 ppm. The IR spectrum of **3** demonstrates an absence of the azide band stretching and the appearance of the band of the triazole cycle at 1462 cm^−1^.

### 2.2. Synthesis of Nido-Carboranyl Cholesterol Derivative with Charge-Compensated Group

Furthermore, compound **2** was used for synthesis of boronated cholesterol as precursor for the preparation of liposomes. The usage of liposomes is the important approach directed to selective delivery of therapeutics into tumors [40,41,42]. The development of selective, non-toxic boron delivery agents that can preferentially deliver a high concentration of boron to the tumor is probably the greatest need for the future progress of BNCT [43]. Due to the high permeability of the walls of blood vessels inside the tumor, stagnant blood flow occurs and lymphatic outflow is disrupted. These changes lead to the EPR (enhanced permeability and retention) effect, due to which macromolecules and nanoparticles such as liposomes penetrate from the bloodstream of the tumor vessel into the intercellular space and accumulate mainly in the tumor tissue [44,45,46]. Cholesterol is the major component of the cell membrane and most liposomal formulations. Therefore, the development of boronated derivatives of cholesterol is an effective approach for the selective delivery of boron clusters into the cancer cells via liposomes. Recently, using “click” reactions we obtained a series of mono-negative charged conjugates of cobalt bis(dicarbollide) with cholesterol [47], conjugates of cobalt/iron of bis(dicarbollide) and cholesterol with similar length spacer but with zwitter-ionic character of target molecule [48] and conjugates of *closo*-dodecaborate dianion with cholesterol [49]. It has been only recently shown that the inclusion of lipophilic boron-containing species in the liposome bilayer provides an attractive method to increase the gross boron content of the liposomes in the formulation [50,51]. In addition, it has been found that PEGylated liposome encapsulating *nido*-carborane by hydrating thin lipid films significantly suppresses tumors in boron neutron capture therapy [52]. 

In the present work, we use the “click” methodology to obtain new conjugate of *nido*-carborane with cholesterol suitable for the preparation of boron-containing liposomes as potential drugs for boron neutron capture therapy of cancer. Usage of *nido*-carborane **2** for synthesis of carborane-cholesterol conjugates leads to zwitter-ionic character of product structure. Its reaction proceeded in a slight excess of alkynyl-3β-cholesterol **4** in the presence of a CuI catalyst and diisopropylethylamine (DIPEA) as a base in ethanol upon prolonged reflux for 8 h to give novel boron conjugate **5** with 85 % yield (Scheme 3).

The ^1^H-NMR spectrum of complex **5** contains a signal for the proton of the triazole group at 8.01 ppm. The characteristics signal of the alkyne C*H*_st_ hydrogen of cholesterol in the conjugate **5** is observed at 5.37 ppm. The spectral characteristics of the C*H* protons of cholesterol are in good agreement with the literature data [53]. The ^13^C-NMR spectrum of **5** exhibits signals for two carbon atoms of the triazole fragment at 145.8 ppm for C*H_triazole_* atom and at 123.1 ppm for *C_triazole_*. In the ^11^H-NMR spectrum, the signal of the *extra-*hydrogen, as expected, is observed approx. at −3.4 ppm. The IR spectrum of compound **5** exhibits absorption bands characteristic of the BH group 2685 cm^−1^ and the triazole ring 1392 cm^−1^.

Based on synthesized compounds the boronated liposomes are planned to prepare in order to deliver boron clusters into a cancer cell for the BNCT experiment.

### 2.3. Synthesis of Zwitter-Ionic Boron-Enriched Cluster Compounds Bearing a 1,2,3-Triazol-Metallacarborane-Nido-Carborane Conjugated Systems 

As was mentioned above, functionalized *nido*-carboranes can be used as building blocks for design and construction of boron-containing compounds for various medical applications [3,4,54]. For example, one of the important requirements of BNCT is the synthesis of structures with a higher content of boron atoms in the molecule than in the clinically used compounds [55,56]. In this contribution, we propose to combine two boron clusters into one molecule: the bis(dicarbollide) cluster serves as a boron-containing substituent providing low toxicity [57,58] and amphiphilicity [59,60] of the molecule and the *nido*-carborane cage serves as a boron-containing base for attachment to molecules by the “click” reaction. Penetration of various substances through biological membranes, their accumulation and retention in cells largely depend on their charge. It is known that positively charged particles have better penetration through biological membranes than negatively changed ones [61,62,63]. It motivated us that the synthesis of such compounds is based on the introduction of two ammonium centers in a spacer: the first one compensating the negative charge of the *nido*-carborane fragment and the second one compensating the charge of the cobalt/iron bis(dicarbollide) moiety. This allows us to double the boron content of the biomolecule as compared to the single cage approach. Moreover, by changing the type and the size of a spacer between these two boron cages, it is possible to control, to some extent, the hydrophilic/hydrophobic balance of the compounds. 

At the first step, cobalt and iron bis(dicarbollide) terminal alkynes with charge-compensated group **6**–**9** were prepared by the cleavage reactions of oxonium derivatives of cobalt/iron bis(dicarbollide) with N,N-dimethylprop-2-yn-1-amine [48,64]. It was found that alkynes prepared from 1,4-dioxane and tetrahydropyran derivatives of cobalt bis(dicarbollide) **6** and **7 [48]** readily undergo “click” reactions with a small excess of azido-derivative of *nido*-carborane **2** to give novel boron conjugates **10** and **11**. High preparative yields (85 %) of the desired products **10** and **11** was achieved using CuI in the presence diisopropylethylamine (DIPEA) as a catalyst and running the reaction for 8 h under reflux. At the same time, usage of the alkyne synthesized from tetrahydropyran derivative of iron bis(dicarbollide) **9** in “click” reaction with **2** leads to dramatically decrease of yield of product **12** (54 %). However, the alkyne synthesized from 1,4-dioxane derivatives of iron bis(dicarbollide) **8** does not react with azido-derivative **2** at all under the same conditions as for the compounds **6**, **7**, **9** and leads to the recovery of the starting materials. It should be noted that a similar difference of behavior of iron and cobalt complexes was observed earlier in the reaction of cobalt/iron bis(dicarbollide) terminal alkynes **6**–**9** with azido-cholesterol [48]. Alkyne **8** had not reacted with azido-cholesterol and had not led to target triazole **12** in contrast to alkynes **6**, **7** and **9**. These results pushed us to carry out the reaction of alkyne **8** with methyl azidoacetate under the same conditions as for compounds **6**, **7** and **9**. However, we did not get the desired result. Employing another amine and/or solvent (e.g., Et_3_N in CH_3_CN, Et_3_N in EtOH, DIPEA in CH_3_CN) did not provide the required material as well. The use of a 3-fold excess of CuI, EtOH as a solvent, DIPEA as a base and running the reaction for 8 h under reflux also did not lead to formation of the target 1,2,3-triazole (Scheme 4). 

Structures of the compounds **10**–**12** were established by ^1^H-, ^11^B- and ^13^C-NMR and IR spectra. In IR spectra of them the absorption bands of BH (2524–2531 cm^-1^) and 1,2,3-triazole (1461–1464 cm^−1^) were observed. In the ^1^H-NMR spectra of the obtained compounds, the characteristics signals of the C*H_triazole_* hydrogens appear in the region at 6.52–8.64 ppm. In the ^13^C-NMR spectra, the signals of the *C*H*_triazole_* carbons for **10**–**12** are observed in the range of 126.8–129.1 ppm, whereas the signals of the *C_triazole_* carbons appear in the range 133.8-135.9 ppm. In IR spectra of **10**–**12** the characteristic bands 1,2,3-triazoles (1461–1464 cm^−1^) were observed.

The compounds obtained are of potential interest for the delivery of boron-enriched drugs in boron neutron capture cancer therapy.

### 2.4. Single-Crystal X-ray Diffraction Studies 

The structure of 9-N_3_CH_2_Me_2_N-*nido*-7,8-C_2_B_9_H_11_
**2** was additionally confirmed by the single crystal X-ray diffraction study (Figure 1). Crystals of **2** suitable of single crystal X-ray analysis were grown from the dichloromethane solution layered with hexane.

Structural features of **2** are expected for this class of compounds (see, for instance [19]) with the only exception of rotation of the substituent at the B9 atom (Figure 1). Namely, the compact conformation of substituent is observed that results in rather short intramolecular contact between the N4 atom of azide function and the H1C *extra*-hydrogen atom of *nido*-carborane cage (3.089 Å with non-normalized B-H bond lengths). A similar contact of the extra hydrogen atom with the acetylene fragment previously found in the crystal of 10-Me_3_SiC≡CCH_2_(Me)S-7,8-C_2_B_9_H_11_ was attributed to the attractive B-H…π interaction which hindered free rotation of substituent in a solution [16]. For **2**, the DFT calculation of the isolated molecule was performed to get more insight into the nature of the H1C…N4 contact. It was found that the conformation of **2** is rather rigid and is influenced by media effects only slightly: the best overlap for non-hydrogen atoms of crystal and isolated molecular structures produces a rather small rmsd value of 0.17 Å (Figure 2). Despite the most deviations are observed for the coordinates of azide fragment, the H1C…N4 contact lengthens insignificantly upon the transition into the isolated state (3.217 Å). According to the analysis of theoretical electron density function ρ(**r**) in terms of the R. Bader’s “Atoms in Molecules” [65] theory, the compact conformation of **2** can be indeed stabilized by intramolecular non-covalent interactions. In particular, there is a bond path connecting the H1C and N4 atoms which can be considered as the privileged channel of interatomic exchange interaction (Figure 3) [66,67]. This bond path is significantly curved in the area of the azide fragment and less curved in the area of the H1C atom. Implying a non-directional character of the interaction the bond path curvature allows to consider at least the π-electrons of the N3-N4 fragment to be involved into the non-covalent bonding with the H1C atom. This is in concordance with the another real-space descriptor of non-covalent interactions based on the reduced density gradient formalism [68]. The distribution of the sign (λ_2_)·ρ(**r**) function (λ_2_ is the intermediate eigenvalue of the ρ(**r**) Hessian) mapped onto the 0.4 isosurface of reduced density gradient (see Figure 3) is characterized by a rather wide region of negative values of sign (λ_2_)·ρ(**r**) which correspond to the concentration of electronic charge between H1C, N4 and N3 atoms. It is interesting to note that the distribution of Bader’s integral atomic charges does not allow to unambiguously treat this interaction as the H-bond (the charges of H1C, N4 and N3 atoms equal to −0.56e, +0.12e and −0.16e, respectively). Finally, according to the ρ(**r**) surface integrals scheme [69], this non-covalent interaction of the *extra*-hydrogen atom is rather strong (2.1 kcal·mol^−1^) and, thus, can be indeed regarded as the important factor which stabilizes the compact conformation of **2** and, probably, relative compounds.

## 3. Materials and Methods

### 3.1. General Methods

9-Cl(CH_2_)_3_Me_2_N-*nido*-7,8-C_2_B_9_H_11_ [19], alkynyl-3β*-*cholesterol [53], cobalt bis(dicarbollides) alkynes **6** and **7** [48], iron bis(dicarbollides) alkynes **8** and **9** [64] were prepared according to the literature. Cholesterol (Fisher Scientific, Loughborough, U.K.), diisopropylethylamine (Carl Roth GmbH, Karlsruhe, Germany), CuI (PANREAC QUIMICA SA, Barcelona, Spain), sodium iodide (Sigma-Aldrich Chemie GmbH, Steinheim, Germany) were used without further purification. DMF, ethanol, CH_3_CN, CH_2_Cl_2_, CH_3_Cl, Et_3_N, phenylacetylene and NaN_3_ were commercially analytical grade reagents. The reaction progress was monitored by thin-layer chromatography (Merck F245 silica gel on aluminum plates). Acros Organics silica gel (0.060–0.200 mm) was used for column chromatography. The NMR spectra at 400.1 MHz (^1^H), 128.4 MHz (^11^B) and 100.0 MHz (^13^C) were recorded with a Bruker Avance-400 spectrometer (Bruker, Karlsruhe-Zurich, Switzerland-Germany). The residual signal of the NMR solvent relative to Me_4_Si was taken as the internal reference for ^1^H- and ^13^C-NMR spectra. ^11^B-NMR spectra were reference using BF_3_*Et_2_O as external standard. Infrared spectra were recorded on an IR Prestige-21 (SHIMADZU, Kyoto, Japan) instrument. High resolution mass spectra (HRMS) were measured on a mictOTOF II (Bruker Daltonic, Bremen, Germany) instrument using electrospray ionization (ESI). The measurements were done in a positive ion mode (interface capillary voltage −4500 V); mass range from *m*/*z* 50 to *m*/*z* 3000; external or internal calibration was done with ESI Tuning Mix, Agilent. A syringe injection was used for solutions in acetonitrile (flow rate 3 µL/min). Nitrogen was applied as a dry gas; interface temperature was set at 180 °C. The electron ionization mass spectra were obtained with a Kratos MS 890 instrument operating in a mass range of *m*/*z* 50–800.

Single crystal X-ray diffraction experiment for 2 was performed using with a Bruker SMART APEX 2 Duo CCD diffractometer [λ(MoKα) = 0.71072Å, ω-scans, 2θ < 60°] (Bruker AXC, Maddison, Wisconsin, USA). The structure was solved by the direct method and refined by the full-matrix least-squares technique against F2 in the anisotropic-isotropic approximation. The hydrogen atoms were found in the difference Fourier synthesis and were further refined in the isotropic approximation without restraints. All calculations were performed using SHELX2018 [70]. The CCDC 2048027 contains the Appendix A. These data can be obtained free of charge via http://www.ccdc.cam.ac.uk/conts/retrieving.html (or from the CCDC, 12 Union Road, Cambridge, CB21EZ, UK; or deposit@ccdc.cam.ac.uk).

The DFT calculations were done in the Gaussian09 program (rev. D01) [71] using the PBE0 functional [72,73] with the Grimme’s D3 dispersion correction [74] and Becke–Jonson damping [75]. The standard def2tzvp basis set and ultrafine integration grids were used. The geometry optimization procedure for the isolated molecule of 2 was performed using standard criteria on displacements and forces. The analysis of the Hessian of potential energy surface for **2** revealed the correspondence of the isolated structure to the energy minimum. The analysis of electron density based on the “Atoms in Molecules” theory together with the reduced density gradient and sign(λ2)·ρ(r) computations were performed in the AIMAll program [76].

### 3.2. Synthesis of 9-N_3_(CH_2_)_3_Me_2_N-nido-7,8-C_2_B_9_H_11_
***2***

Compound **1** (1.00 g, 0.0034 mol) was dissolved in 50 mL of DMF and NaN_3_ (0.44 g, 0.0068 mol) and anhydrous NaI (0.05 g, 0.0003 mol) were added. The reaction mixture was heated at 50 °C for 7 days. Then the reaction mixture was cooled to room temperature, and 50 mL of H_2_O was added. The formed precipitate was filtered and dried in the air, then it was purified on silica column by using CH_2_Cl_2_-CH_3_CN (5:1) as eluent to give white solid of **2** (0.80 g, yield 90%). ^1^H-NMR (400 MHz, methanol-*d*_4_): δ 3.50 (2H, m, CH_2_N_3_), 3.37 (2H, s, Me_2_NCH_2_), 2.99 (3H, s, NMe), 2.93 (3H, s, NMe), 2.59 (1H, s, CH_carb_), 2.15 (2H, s, CH_2_CH_2_CH_2_), 1.87 (1H, s, CH_carb_), −3.42 (1H, br. s, H_extra_) ppm; ^11^B-NMR (128 MHz, methanol-*d*_4_): δ 5.2 (1B, s), −5.4 (1B, d, *J* = 139), −16.9 (2B, unsolved d,), −17.5 (1B, unsolved d,) −25.0 (1B, d, *J* = 147), −27.4 (1B, d, *J* = 143), −32.2 (1B, d, *J* = 135 Hz), −38.7 (1B, d, *J* = 145 Hz) ppm; ^13^C-NMR (101 MHz, methanol-*d*_4_) δ 65.2 (CH_2_N_3_), 52.7 (NMe), 50.4 (NMe), 48.5 (Me_2_NCH_2_), 46.3 (br., C_carb_), 33.7 (br., C_carb_), 23.6 (CH_2_CH_2_CH_2_) ppm. IR (solid): ν˜ = 2537 cm^−1^ (BH), 2075 cm^−1^ (N_3_), HRMS-ESI^+^
*m*/*z* for [C_7_H_23_B_9_N_4_ + NH_4_]^+^ calcd 300.2435, found 300.2434. Crystallograhic data: crystals of **2** (C_7_H_23_B_9_N_4_, M = 260.58) are monoclinic, space group P2_1_/c, at 120K: a = 6.7746(5), b = 26.5914(18), c = 8.8809(6), β = 111.415(2), V = 1489.41(18) Å^3^, Z = 4 (Z’ = 1), d_calc_ = 1.162 g·cm^−3^. wR_2_ and GOF converged to 0.1292 and 0.991 for all independent reflections; R_1_ = 0.0461 was calculated for 3133 observed reflections with I > 2σ(I)).

#### General Procedure for the Synthesis of the Compounds **3**, **5**, **10**–**12**

A mixture of compound **2** (1,2 eq.), alkene (1 eq.), diisopropylethylamine (0,5–1 mL) and CuI (0,1 eq.) in 10–20 mL ethanol was heated under reflux for 8 h. Then the reaction mixture was cooled to room temperature and was passed through ca. 2–3 cm of silica. Then solvent was removed in vacuo. The crude product was purified on a silica column using CH_2_Cl_2_-CH_3_CN as an eluent to give the desired products **3, 5, 10**–**12**.

### 3.3. Synthesis of 9-(Ph)C-CH-N_3_(CH_2_)_3_Me_2_N-nido-7,8-C_2_B_9_H_11_
***3***

Prepared from compound **2** (0.10 g, 0.38 mmol), phenylacetylene (0.035 mL, 0.033 g, 0.32 mmol), diisopropylethylamine (1 mL, 0.74 g, 5.73 mmol) and CuI (0.006 g, 0.03 mmol) in 20 mL of ethanol. The product was obtained as a a white solid of **3** (0.10 g, yield 85%). ^1^H-NMR (400 MHz, DMSO-*d*_6_) δ 8.62, (1H, s, CC*H*N_3_), 7.84 (2H, d, Ph, *J* = 7.6 Hz), 7.46 (2H, t, Ph, *J* = 7.5 Hz), 7.34 (1H, t, Ph, *J* = 7.4 Hz), 4.54 (2H, t, -C*H_2_*-CCHN_3_, *J* = 6.9 Hz,), 3.34 (2H, t, C*H_2_*NMe_2_, *J* = 8.5 Hz), 2.88 (6H, s, N*Me_2_*), 2.77 (1H, s, C*H_carb_*), 2.41 (2H, m, CH_2_-C*H_2_*-CH_2_), 1.91 (1H, s, C*H_carb_*), −3.45 (1H, br. s, *H_extra_*) ppm; ^11^B-NMR (128 MHz, DMSO-*d*_6_) δ 5.7 (1B, s), −6.0 (1B, unsolved d), −18.0 (3B, unsolved d), −25.1 (1B, unsolved d), −27.5 (1B, d, *J* = 117 Hz), −32.7 (1B, unsolved d), −38.7 (1B, d, *J* = 129 Hz) ppm; ^13^C-NMR (101 MHz, DMSO-*d*_6_): δ 146.9 (*C*CHN_3_), 131.1 (Ph), 129.4 (Ph), 128.4 (Ph), 125.6 (Ph), 122.0 (C*C*HN_3_), 64.4 (*C*H_2_-NMe_2_), 53.2 (N*Me_2_*), 51.4 (N*Me_2_*), 47.5 (*C*H_2_-CCHN_3_), 34.1 (C_carb_), 25.0 (CH_2_-*C*H_2_-CH_2_) ppm. IR (solid): ν˜ = 2546 cm^−1^ (BH), 1462 cm^−1^ (triazole), HRMS-ESI^+^
*m/z* for [C_15_H_29_B_9_N_4_ + H]^+^ calcd 363.3377, found 363.3386.

### 3.4. Synthesis of 9-3β-Chol-O(CH_2_)C-CH-N_3_(CH_2_)_3_Me_2_N-nido-7,8-C_2_B_9_H_11_
***5***

Prepared from compound **2** (0.072 g, 0.28 mmol), alkynyl-cholesterol **4** (0.10 g, 0.23 mmol), diisopropylethylamine (1 mL, 0.74 g, 5.73 mmol) and CuI (0.002 g, 0.03 mmol) in 20 mL of ethanol. The product was obtained as a white solid of **5** (0.10 g, yield 85 %). ^1^H-NMR (400 MHz, acetone-*d*_6_) δ 8.01 (1H, s, CC*H*N_3_), 5.37 (1H, m, C_st_(6)H), 4.64 (2H, s, -C*H_2_*-CCHN_3_), 4.63 (2H, m, CCHN_3_-C*H_2_*), 3.54 (2H, m, C*H_2_*NMe_2_), 3.31 (1H, m, C_st_(3)H), 3.08 (3H, s, N*Me*_2_), 3.07 (3H, s, N*Me*_2_), 2.64 (3H, m, C*H_2_*, C*H_carb_*), 2.42 (1H, m, C_st_(H)), 2.19 (1H, m, C_st_(H)), 1.88 (6H, m, C_st_(H), C*H_carb_*), 1.33 (20H, m, C_st_(H)), 1.04 (3H, s, C_st_(19)H_3_), 0.97 (3H, d, *J* = 6.4, C_st_(21)H_3_,), 0.90 (3H, d, *J* = 1.4, C_st_(26)H_3_), 0.88 (3H, d, *J* = 1.4, C_st_(27)H_3_), 0.74 (3H, s, C_st_(18)H_3_), −3.39 (1H, br. s., *H_extra_*) ppm; ^11^B (128 MHz, acetone-*d*_6_) δ 5.4 (1B, s), −5.5 (1B, d, *J* = 141), −17.2 (2B, d, *J* = 180), −18.9 (1B, d, *J* = 120), −24.9 (1B, d, *J* = 149), −27.3 (1B, d, *J* = 142), −32.3 (1B, d, *J* = 166), −38.6 (1B, d, *J* = 144) ppm; ^13^C-NMR (101 MHz, acetone-*d*_6_) δ 145.8 (*C*CHN_3_), 140.8 (C_st_(5)), 123.1 (C*CH*N_3_), 121.3 (C_st_(6)), 78.2 (C_st_(3)), 64.7 (CCHN_3_-*CH_2_*), 61.1 (*CH_2_*-CCHN_3_), 56.7 (C_st_(14)),56.2 (C_st_(17)), 52.8 (N*Me*_2_), 51.1 (N*Me*_2_), 50.3 (C_st_(9)), 47.1 (*C*H_2_NMe_2_), 42.2 (C_st_(4)), 39.8 (C_carb_), 39.4 (C_st_(12)), 39.0 (C_st_(13)), 37.1 (C_st_(24)), 36.7 (C_st_(1)), 36.1 (C_st_(10)), 35.7 (C_st_(22)), 34.0 (C_carb_), 31.9 (C_st_(20)), 31.8 (C_st_(8)), 28.2 (C_st_(2)), 28.0 (C_st_(7)), 27.8 (C_st_(16)), 25.0 (C_st_(25)), 24.1 (C_st_(15)), 23.6 (CH_2_), 22.2 (C_st_(23)), 22.0 (C_st_(26), C_st_(27)), 20.9 (C_st_(11)), 18.9 (C_st_(19)), 18.3 (C_st_(21)), 11.4 (C_st_(18)) ppm. IR (solid): ν˜ = 2685 cm^−1^ (BH), 1392 cm^−1^ (triazole), HRMS-ESI^+^
*m/z* for [C_37_H_71_B_9_N_4_O + H]^+^ calcd 685.6614, found: 685.6627.

### 3.5. Synthesis of 9-[(8′-Me_2_N(CH_2_)_3_N_3_CCH(CH_2_)NMe_2_(CH_2_CH_2_O)_2_-1′,2′-C_2_B_9_H_10_)-3′,3″-Co(1″,2″-C_2_B_9_H_11_)]-nido-7,8-C_2_B_9_H_11_
***10***

Prepared from compound **2** (0.06 g, 0.24 mmol), compound **6** (0.10 g, 0.20 mmol), diisopropylethylamine (1 mL, 0.74 g, 5.73 mmol) and CuI (0.004 g, 0.02 mmol) in 20 mL of ethanol. The product was obtained as an orange solid of **10** (0.13 g, yield 85%). ^1^H-NMR (400 MHz, acetone-*d*_6_) δ 8.57 (1H, s, CC*H*N_3_), 4.97 (2H, s, Me_2_N-C*H_2_*-CCHN_3_), 4.74 (2H, q, *J* = 6.3 Hz, CCHN_3_-C*H_2_*), 4.13 (4H, m, B-OC*H_2_*, C*H_carb_*), 4.00 (2H, br. s, C*H_carb_*), 3.74 (2H, m, B-OC*H_2_*), 3.66 (4H, m, 2[C*H_2_*NMe_2_]), 3.55 (2H, m, B-OC*H_2_*), 3.40 (6H, s, N*Me_2_*), 3.08 (3H, s, N*Me*_2_), 3.06 (3H, s, N*Me*_2_), 2.68 (3H, s, C*H_2_*, C*H_carb_*), 1.89 (1H, s, C*H_carb_*), −3.45 (1H, br. s., *H_extra_*) ppm; ^11^B-NMR (128 MHz, acetone-*d*_6_) δ 24.5 (1B, s, B-O), 5.3 (1B, s, B-N), 0.2 (1B, d, *J =* 138), −2.8 (1B, d, *J =* 143), −6.9 (2B, d, *J =* 178), −6.9 (4B, d, *J =* 137), −9.1 (2B, unsolved d), −17.2 (3B, d, *J =* 148), −18.9 (2B, unsolved d), −20.2 (2B, d, *J =* 163), −22.4 (1B, unsolved d), −24.9 (1B, d, *J =* 150), −27.4 (1B, *J =* 140), −28.9 (1B, unsolved d), −32.4 (1B, d, *J =* 135), −38.6 (1B, d, *J =* 144) ppm; ^13^C-NMR (101 MHz, acetone-*d*_6_) δ 135.9 (*C*CHN_3_), 129.1 (C*C*HN_3_), 72.2 (O*C*H_2_), 68.8 (O*C*H_2_), 64.8 (*C*H_2_NMe_2_), 64.5 (*C*H_2_-N_3_CCH), 62.7 (*C_carb_*), 59.5 (*C_carb_*), 52.9 (N*Me*_2_), 52.2 (N*Me*_2_), 51.5 (N*Me_2_*), 47.5 (B-Me_2_N*C*H_2_), 46.5 (*C_carb_*), 34.0 (*C_carb_*), 25.1 (*C*H_2_) ppm. IR (solid): ν˜ = 2530 cm^−1^ (BH), 1463 cm^−1^ (triazole), HRMS-ESI^+^
*m/z* for [C_20_H_61_B_27_CoN_5_O_2_ + H]^+^ calcd 755.6936, found: 755.6929.

### 3.6. Synthesis of 9-[(8′-Me_2_N(CH_2_)_3_N_3_CCH(CH_2_)NMe_2_(CH_2_)_5_-1′,2′-C_2_B_9_H_10_)-3′,3″-Co(1″,2″-C_2_B_9_H_11_)]-nido-7,8-C_2_B_9_H_11_
***11***

Prepared from compound **2** (0.06 g, 0.24 mmol), compound **7** (0.10 g, 0.20 mmol), diisopropylethylamine (1 mL, 0.79 g, 6.10 mmol) and CuI (0.004 g, 0.02 mmol) in 20 mL of ethanol. The product was obtained as an orange solid of **11** (0.13 g, yield 85 %). ^1^H-NMR (400 MHz, acetone-*d*_6_) δ 8.64 (1H, s, CC*H*N_3_), 4.91 (2H, s, Me_2_N-C*H_2_*-CCHN_3_), 4.76 (2H, q, *J* = 6.4 Hz, CCHN_3_-C*H_2_*), 4.20 (2H, br. s, C*H_carb_*), 4.10 (2H, br. s, C*H_carb_*), 3.56 (2H, m, B-OC*H_2_*), 3.49 (4H, m, 2[C*H_2_*NMe_2_]), 3.33 (6H, s, N*Me_2_*), 3.09 (3H, s, N*Me*_2_), 3.08 (3H, s, N*Me*_2_), 2.71 (3H, s, C*H_2_*, C*H_carb_*), 1.88 (1H, s, C*H_carb_*), 1.57 (2H, m, CH_2_C*H_2_*CH_2_), 1.48 (4H, m, C*H_2_*CH_2_C*H_2_*), −3.45 (1H, br. s., *H_extra_*) ppm; ^11^B-NMR (128 MHz, acetone-*d*_6_) δ 23.8 (1B, s, B-O), 5.1 (1B, s, B-N), 0.0 (1B, d, *J =* 138), −2.7 (1B, d, *J =* 150), 4.6 (1B, unsolved d), −5.1 (1B, unsolved d), −7.7 (8B, m), −17.3 (3B, d, *J =* 134), −18.8 (2B, unsolved d), −20.3 (2B, d, *J =* 153), −22.4 (1B, unsolved d), −25.0 (1B, d, 171), −27.4 (1B, d, *J =* 151), −28.8 (1B, d, *J =* 143), −32.4 (1B, d, *J =* 118), −38.6 (1B, d, *J =* 171) ppm; ^13^C-NMR (101 MHz, acetone-*d*_6_) δ 135.5 (*C*CHN_3_), 128.3 (C*C*HN_3_), 68.2 (O*C*H_2_), 64.4 (*C*H_2_NMe_2_), 64.4 (*C*H_2_-N_3_CCH), 58.1 (*C_carb_*), 53.2 (*C_carb_*), 53.0 (N*Me*_2_), 51.6 (N*Me*_2_), 50.3 (N*Me_2_*), 47.6 (B-Me_2_N*C*H_2_), 46.4 (*C_carb_*), 34.0 (*C_carb_*), 25.1 (*C*H_2_), 23.1 (*C*H_2_), 22.1 (*C*H_2_) ppm. IR (solid): ν˜ = 2531 cm^−1^ (BH), 1461 cm^−1^ (triazole), HRMS-ESI^+^
*m/z* for [C_21_H_63_B_27_CoN_5_O + H]^+^ calcd 753.7143, found 753.7138.

### 3.7. Synthesis of 9-[(8′-Me_2_N(CH_2_)_3_N_3_CCH(CH_2_)NMe_2_(CH_2_)_5_-1′,2′-C_2_B_9_H_10_)-3′,3″-Fe(1″,2″-C_2_B_9_H_11_)]-nido-7,8-C_2_B_9_H_11_
***12***

Prepared from compound **2** (0.03 g, 0.12 mmol), compound **9** (0.05 g, 0.10 mmol), diisopropylethylamine (0.5 mL, 0.37 g, 2.87 mmol) and CuI (0.002 g, 0.01 mmol) in 20 mL of ethanol. The product was obtained as an orange solid of **12** (0.04 g, yield 54%). ^1^H-NMR (400 MHz, acetone-*d*_6_) δ 72.2 (br. s, BH), 67.38 (br. s, BH), 45.15 (br. s, BH), 43.16 (br. s, BH), 37.74 (br. s, CH_carb_), 6.52 (1H, s, CC*H*N_3_), 4.27 (2H, br s, Me_2_N-C*H_2_*-CCHN_3_), 3.15 (2H, t, *J =* 7.7, C*H_2_*NMe_2_), 2.96 (3H, s, N*Me*_2_), 2.89 (3H, s, N*Me*_2_), 2.79 (6H, s, N*Me_2_*), 2.53 (1H, s, C*H_carb_*), 2.28 (2H, m, C*H_2_*), 1.64 (1H, br s, C*H_carb_*), −3.88 (1H, br s, *H_extra_*), −5.25 (2H, m, OC*H_2_*), −10.22 (2H, br s, C*H_2_*), −11.26 (2H, br s, C*H_2_*), −11.85 (2H, br s, C*H_2_*) ppm; ^11^B-NMR (128 MHz, acetone-*d*_6_) δ 118.8, 100.8, 36.9, 27.1, 5.0, −1.8, −6.1, −17.5, −19.1, −25.2, −27.8, −32.8, −38.9, −370.6, −382.7, −448.1, −486.7 ppm; ^13^C-NMR (101 MHz, acetone-*d*_6_) δ 133.8 (*C*CHN_3_), 126.8 (C*C*HN_3_), 64.0 (*C*H_2_-N_3_CCH), 59.5 (*C*H_2_NMe_2_), 56.0 (*C*H_2_NMe_2_), 52.7 (N*Me*_2_), 51.4 (N*Me*_2_), 48.1 (B-Me_2_N*C*H_2_), 47.1 (*C_carb_*), 35.9 (*C_carb_*), 24.5 (*C*H_2_), 14.9 (*C*H_2_), 14.5 (*C*H_2_), 10.8 (B-O*C*H_2_) ppm. IR (solid): ν˜ = 2524 cm^−1^ (BH), 1464 cm^−1^ (triazole), HRMS-ESI^+^
*m/z* for [C_21_H_63_B_27_FeN_5_O + H]^+^ calcd 750.7161, found 750.7154.

## 4. Conclusions

In this work we prepared and characterized novel *nido*-carboranyl azide 9-N_3_(CH_2_)_3_Me_2_N-*nido*-7,8-C_2_B_9_H_11_ derived from the reaction of 9-Cl(CH_2_)_3_Me_2_N-*nido*-7,8-C_2_B_9_H_11_ with NaN_3_ in the presence of NaI as a catalyst in strong conditions. The possibility of using “click” approach in regard to the obtained compound was demonstrated on the reaction of 9-N_3_(CH_2_)_3_Me_2_N-*nido*-7,8-C_2_B_9_H_11_ with phenylacetylene. We also studied the behaviour of *nido*-carboranyl azide in the copper(I)-catalyzed azide-alkyne cycloaddition reaction with alkynyl-cholesterol and obtained new the *nido*-carborane-cholesterol conjugate with charge-compensated group in a linker in a good yield. Based on the synthesized compound, the boronated liposomes are planned for preparation in order to deliver boron clusters into a cancer cell for a BNCT experiment. It should be concluded that ”click“ reactions between 9-N_3_(CH_2_)_3_Me_2_N-*nido*-7,8-C_2_B_9_H_11_ and cobalt/iron bis(dicarbollide) terminal alkynes lead to novel zwitter-ionic boron-enriched cluster compounds bearing a 1,2,3-triazol-metallacarborane-carborane conjugated system. By changing the type and the size of a spacer between these two boron cages, it is possible to control the hydrophilic/hydrophobic balance of the compounds. Prepared conjugates contain a large amount of boron atom in the biomolecule potentially can be used for boron neutron capture therapy of cancer (BNCT).

The solid-state molecular structure of the novel *nido*-carboranyl azide 9-N_3_(CH_2_)_3_Me_2_N-*nido*-7,8-C_2_B_9_H_11_ was determined by single-crystal X-ray diffraction studies. Structural feature of *nido*-carboranyl azide is rather rigid and compact conformation of substituent is observed that results in rather short intramolecular contact between the N4 atom of azide function and the H1C *extra*-hydrogen atom of *nido*-carborane cage (3.089 Å with non-normalized B-H bond lengths).

## Data Availability

The data presented in this study are available in Appendix A.

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
