# Peer review of "Synthesis and Structure of *Nido*-Carboranyl Azide and Its “Click” Reactions"

_molecules, 2021, doi:10.3390/molecules26030530_

Round 1

Reviewer 1 Report

The manuscript is appropriate for publication in Molecules, but a number of technical issues should be addressed prior to publication:

1) Materials and Methods should appear BEFORE Results and Discussion.  For example, the specification of the method in the DFT calculations should appear before any discussion of results from said calculations.

2) Strange inserts like “click” on line 3 and “Bookmark not defined” on line 248 need to be corrected.

3) The acronym BNCT occurs in the abstract without specification of what BNCT stands for.

4) The paper has significant chemical AND biological implications, but the abstract seems to focus solely on the chemistry.  The abstract should be rewritten to better convey the biological importance of the paper.

Author Response

Thank you for your recommendations

Point 1: 1) Materials and Methods should appear BEFORE Results and Discussion. For example, the specification of the method in the DFT calculations should appear before any discussion of results from said calculations.

Response 1: The journal rules do not impose hard rules on the structure of the manuscript. Therefore, the authors consider this arrangement of chapters to be correct from the point of view of the perception of the text. According to Journal rules details of chemical and X-ray experiments as well as DFT calculations presented in "Experimental section".

Point 2: Strange inserts like “click” on line 3 and “Bookmark not defined” on line 248 need to be corrected.

Response 2: Corrected. The authors corrected the link 54 on line 265. On line 3 authors corrected inserts for the word “click”.

Point 3: The acronym BNCT occurs in the abstract without specification of what BNCT stands for.

Response 3: Corrected. The authors clarified the acronym BNСT on line 18 «(Boron Neutron Capture Therapy)».

Point 4: The paper has significant chemical AND biological implications, but the abstract seems to focus solely on the chemistry. The abstract should be rewritten to better convey the biological importance of the paper.

Response 4: Corrected. The authors rewrote the abstract by adding the biological potential application of boron-enriched cluster compounds on line 21-22 «Prepared conjugates contain a large amount of boron atom in the biomolecule and potentially can be used for BNCT». The biological application of nido-carborane-cholesterol conjugate was indicated in the abstract on line 17-18.

With kind regards,

Anna Druzina (on behalf of the authors)

Reviewer 2 Report

This manuscript presents the synthesis of nido-carboranyl azide 9-N3(CH2)3Me2N-nido-7,8-C2B9H11 from the reaction of 9-Cl(CH2)3Me2N-nido-7,8-C2B9H11 and NaN3 in high yield. The resulted azide product can be efficiently used for the preparation of novel zwitter-ionic boron-enriched cluster compounds through click reactions. These boron-containing conjugates can be potentially used for boron neutron capture therapy of cancer.

Overall, this manuscript develops the method to access the poly-boron conjugates using the nido-carboranyl azide as the starting material through click reactions. I strongly recommend for publication.

Author Response

We are grateful for your attention to our work.

With kind regards,

Anna Druzina (on behalf of the authors)

Reviewer 3 Report

Druzina and coworkers reported nido-carboranyl azide and used it for copper-catalyzed azide-alkyne click reaction. 

  1. The most ideal scenario for applying click reactions are ambient and fast reactions. The reactions reported in this work all required reflux condition in ethanol. Is there any possibility to make this reaction "greener" for future applications, and what are the possible applications of these newly synthesized nido-carboranyl compounds? Please address these two questions in the Discussion section.
  2. Several references on click reactions are suggested to be included: DOI: 10.1002/9780470559277.ch110148; DOI: 10.1016/j.ccr.2011.06.028; DOI: 10.1002/cpch.85; DOI: 10.1016/j.trechm.2020.03.007; DOI: 10.1021/acs.chemrev.5b00408.
  3. Line 248, typo. 

Author Response

Thank you for your recommendations.

Point 1: The most ideal scenario for applying click reactions are ambient and fast reactions. The reactions reported in this work all required reflux condition in ethanol. Is there any possibility to make this reaction "greener" for future applications, and what are the possible applications of these newly synthesized nido-carboranyl compounds? Please address these two questions in the Discussion section.

Response 1: Corrected. The authors responded to the correct comments of the reviewer and added the answers in the Discussion section.

The possible applications of newly synthesized nido-carboranyl compounds are mentioned on line 157-158 «Based on synthesized compounds the boronated liposomes are planned to prepare in order to deliver boron clusters into a cancer cell for BNCT experiment» and on line 215-216 «The compounds obtained are of potential interest for the delivery of boron-enriched drugs in boron neutron capture therapy of cancer».

The possibility to make this reaction "greener" for future applications is discussed on line 81-82 «It was isolated as a white non-hydroscopic solid soluble in common organic solvents like СH2Cl2, CH3CN, alcohols and non-soluble in hydrocarbons and water» and on line 95-104 «Conditions of «click» reactions in the preparation of various boron-containing biomolecules vary particularly wide. Earlier «click» reactions have been successfully used for the synthesis of conjugates of bis(dicarbollide) metallacarboranes and nido-carborane with thymidine [14]. The synthesis was carried out in a mixture of tert-butanol/water (1:1) at ambient temperature using copper(II) sulfate pentahydrate with potassium ascorbate as a catalyst. The same reaction for synthesis of conjugate dodecaborate dianion with thymidine proceed in CH3CN at ambient temperature with copper(II) sulfate pentahydrate with sodium ascorbate [39]. Conjugates of chlorine e6 with cobalt bis(dicarbollide) anion or closo-dodecaborate dianion were obtained using CuI and Et3N in acetonitrile at ambient temperature [35]. Series of 1,2,3-triazoles bearing closo-dodecaborate fragment was obtained using CuI as a catalyst and Et3N as a base under reflux in ethanol [40]».

.

Point 2: Several references on click reactions are suggested to be included: DOI: 10.1002/9780470559277.ch110148; DOI: 10.1016/j.ccr.2011.06.028; DOI: 10.1002/cpch.85; DOI: 10.1016/j.trechm.2020.03.007; DOI: 10.1021/acs.chemrev.5b00408.

Response 2: Corrected. Authors included these references on click reactions.

Lines 55 and 490-499.

Point 3: Line 248, typo.

Response 3: Corrected. The authors corrected the link 54 on line 265

With kind regards,

Anna Druzina (on behalf of the authors)
